# Beyond Game Theory Optimal: Profit-Maximizing Poker Agents for No-Limit Hold'em

## Abstract

Game theory has grown into a major field over the past few decades, and poker has long served as one of its key case studies. Game-Theory-Optimal (GTO) provides strategies to avoid loss in poker, but pure GTO does not guarantee maximum profit. To this end, we aim to develop a model that outperforms GTO strategies to maximize profit in No Limit Hold'em, in heads-up (two-player) and multi-way (more than two-player) situations. Our model finds the GTO foundation and goes further to exploit opponents. The model first navigates toward many simulated poker hands against itself and keeps adjusting its decisions until no action can reliably beat it, creating a strong baseline that is close to the theoretical best strategy. Then, it adapts by observing opponent behavior and adjusting its strategy to capture extra value accordingly. Our results indicate that Monte-Carlo Counterfactual Regret Minimization (CFR) performs best in heads-up situations and CFR remains the strongest method in most multi-way situations. By combining the defensive strength of GTO with real-time exploitation, our approach aims to show how poker agents can move from merely not losing to consistently winning against diverse opponents.

## 1 Introduction

Poker has evolved from a niche pastime into a global mind sport and a natural laboratory for studying decision making under imperfect information. Online platforms and televised tournaments now generate billions of recorded hands each year, enabling quantitative analyses of risk, bluffing, and opponent modelling. No-Limit Texas Hold'em (NLHE) is especially prominent: state-of-the-art solvers compute game-theory-optimal (GTO) strategies—approximate Nash equilibria that guarantee long-term unexploitable play. Yet in real games, opponents consistently deviate from equilibrium. As a long-time poker enthusiast and researcher in multi-agent learning, the largest profits come from detecting and exploiting such deviations in real time. Current poker AIs remain either purely GTO (safe but conservative) or trained on static population tendencies (often slow to adjust), leaving a gap for dynamic, table-specific exploitation with provable safety. [1, 2, 3]

Building on these observations, we investigate algorithmic approaches that can both approximate game-theoretic optimal play and adapt to changing opponents. A central concept is *counterfactual regret*, which quantifies how much better a player could have done in hindsight by choosing a different action in a given decision situation. Counterfactual Regret Minimization (CFR) repeatedly simulates play, measures these regrets, and adjusts its strategy to minimize them, a process that provably converges to a Nash equilibrium in *two-player zero-sum finite* extensive-form games and, in practice, on abstracted NLHE where card and action spaces are discretized.. [4, 5, 6] Monte-Carlo CFR (MCCFR) improves computational efficiency by sampling single trajectories instead of traversing the full game tree. [7, 8, 9]

Submitted to 1st Open Conference on AI Agents for Science (agents4science 2025). Do not distribute.

Deep CFR extends this framework with neural networks that generalize across large state spaces, while Neural Fictitious Self-Play (NFSP) blends reinforcement learning and supervised learning to maintain an average policy that also approaches equilibrium. These algorithms define today's standard toolkit for large-scale imperfect-information games, providing the backbone for most competitive poker AIs. [10, 11, 12]

Despite their success, purely equilibrium-seeking methods are slow to exploit opponents who deviate from optimal play, and population-trained agents often fail to adapt when table dynamics differ from historical data. This motivates our proposed approach: an adaptive model that learns GTO behaviour from self-play while continuously tracking opponents' tendencies and shifting strategy in real time to capture excess value, all while maintaining provable safety against counter-exploitation. [13]

In summary, our contributions are as follows:

1. Evaluate whether leading algorithms can converge to pure game-theory-optimal (GTO) strategy in No-Limit Hold'em.

2. Examine their ability to construct adaptive GTO-like strategies that respond to specific opponent behaviors.

3. Extend this adaptive framework to multi-player (multiway) scenarios to test scalability beyond heads-up plays.

This research was conducted in full compliance with the Agents4Science Code of Ethics. It involves only synthetic poker decision states and contains no human or sensitive data, follows principles of scientific integrity and reproducibility, and poses no foreseeable harm to people, animals, or the environment.

## 2 Related Work

**Models.** **Counterfactual Regret Minimization (CFR)** is a foundational algorithm for solving large two-player zero-sum imperfect-information games. It iteratively traverses the game tree, computing counterfactual regrets at every information set and adjusting action probabilities to minimize those regrets, and the average of the successive strategies provably converges to a Nash equilibrium. Since its introduction, CFR has become the standard baseline for equilibrium computation in heads-up no-limit Texas Hold'em and related poker games. [4, 5, 14, 6] **Monte-Carlo CFR (MCCFR)** improves the scalability of CFR by sampling single trajectories rather than exhaustively traversing the full game tree. By replacing deterministic updates with stochastic sampling, MCCFR dramatically reduces memory and runtime while retaining theoretical convergence guarantees, and it is widely used when exact traversal is impractical, serving as the default method for building strong poker agents at reasonable computational cost. [7, 8, 9] **Deep CFR** further extends this framework by replacing the tabular regret and strategy tables of CFR with neural function approximators: a regret network predicts counterfactual regrets and a policy network predicts the average strategy, enabling generalization across similar states and scalability to much larger action and information spaces. This deep-learning extension has been used to train near-equilibrium strategies in very large no-limit Hold'em subgames and other imperfect-information domains. [10, 15] **Neural Fictitious Self-Play (NFSP)** combines reinforcement learning with supervised learning to approximate fictitious play in large games. A reinforcement-learning component continually improves a best-response policy, while a supervised component maintains an average policy that approaches equilibrium, offering a fully online, self-play training regime that has been demonstrated on full-scale poker as well as other multi-agent settings. [11, 12, 16]

**Random Policy.** A uniform random policy serves as a non-strategic baseline: at each decision point it samples among legal actions with equal probability. Although it has no convergence guarantees and performs poorly in practice, it provides a lower bound for evaluating how much structure the learning algorithms extract from the game.

Together these prior methods define the standard algorithmic landscape for equilibrium approximation and adaptive play in large imperfect-information games. Our work builds on this literature by benchmarking all four learning algorithms and a random baseline within a unified experimental framework and by quantifying their distance to a strong MCCFR-trained GTO proxy.

# 3 No Limit Texas Hold'em (NLHE) Basics

No Limit Texas Hold'em (NLHE) is the most widely played variant of poker in both live cash games and tournaments. Each player is dealt two private hole cards, followed by five community cards dealt face up in three stages: the flop (three cards), turn (one card), and river (one card). Betting rounds occur after the hole cards and after each community stage. "No Limit" means a player may wager any amount of their remaining chips at any time, from the minimum bet to an all-in shove, making stack depth and bet sizing central to strategic decision making.

The goal is to form the best possible five-card hand using any combination of the two hole cards and the five community cards, or to win the pot uncontested through betting. For example, holding A♡K♡ on a board of Q♡J♣5♢10♡2♠ yields a Broadway straight (Ten through Ace). Another scenario might involve pocket pairs such as 9♠9♣ on a board of 9♡4♢4♣K♠2♡, giving a full house (nines over fours).

Each hand follows a fixed betting sequence. The two players to the left of the dealer post the small and big blinds to seed the pot. Pre-flop action begins with the player to the left of the big blind and proceeds clockwise. After the flop, turn, and river, players may check, bet, call, raise, or fold, depending on prior action. The combination of unrestricted bet sizes and multiple betting rounds rewards players who can balance strong value hands with well-timed bluffs, calculate pot odds and implied odds, and read opponents' likely ranges.

## 3.1 Keyword Definitions

**Basic Actions: Fold, Bet, Call, Raise, and All-in.** These fundamental betting actions govern how chips move during each round of No Limit Texas Hold'em. A *fold* means discarding one's cards and forfeiting any claim to the current pot, immediately ending the player's participation in the hand. A *bet* is the first voluntary wager made on a given street (pre-flop, flop, turn, or river). A *call* matches the current bet to stay in the hand without increasing the size of the pot. A *raise* increases the wager beyond the existing bet, applying pressure to opponents and potentially extracting more value from strong holdings. An *all-in* occurs when a player wagers all remaining chips, creating a side pot if other players have more chips than the all-in player.

**Flop, Turn, and River.** In Texas Hold'em, the *flop* is the first set of three community cards dealt face up after the initial pre-flop betting round, providing most of the shared information that shapes each player's strategy. The *turn* is the fourth community card revealed, adding further possibilities for draws and made hands. Finally, the *river* is the fifth and last community card, completing the board and setting the stage for the final round of betting before a potential showdown.

**Board texture.** The overall arrangement of community cards—called the *board texture*—profoundly shapes betting decisions and equity distribution. Rather than listing every category here, we refer to Table 1, which details representative textures and their strategic implications. In play, a "dry" flop might encourage small continuation bets, while more connected or flush-prone textures create volatile, draw-heavy situations that invite larger bets and frequent raises.

**Game Theory Optimal (GTO) and Exploitable Play** *GTO* (Game Theory Optimal) refers to a balanced poker strategy that cannot be profitably exploited, because it mixes actions in mathematically optimal proportions against any opponent. An intuitive analogy is the game of rock–paper–scissors: a pure GTO approach throws each option exactly 33% of the time so that no counter-strategy gains an edge. By contrast, an *exploitable* strategy contains predictable weaknesses that skilled opponents can identify and profit from; exploitative play is like increasing the frequency of rock when an opponent consistently throws scissors. While pure GTO play minimizes long-term losses even against perfect opposition, many successful players intentionally deviate from GTO to exploit specific tendencies of weaker opponents when the expected value gain outweighs the risk of being countered.

For a complete glossary of the following terms including, *odds, bluff, pot, range, poker agent, read, equity, winning hand, hand ranks* can be found in the Appendix A..

Table 1: Board texture categories for texture sampling in poker simulations.

| Category | Definition | Strategic impact |
|---|---|---|
| `dry` | Flop with few coordinated draws, e.g., K♣7♢2♠; cards are well spaced and mostly rainbow. | Limited straight/flush potential; leads to smaller continuation bets and fewer bluffs. |
| `paired` | One rank appears twice, e.g., 9♠9♢4♡. | Trips/full-house possibilities dominate; incentives for slow-playing or pot control. |
| `two_tone` | Exactly two suits present, creating a flush draw, e.g., Q♠8♠3♢. | Flush-draw equity encourages larger pots and semi-bluffs. |
| `monotone` | All three flop cards share the same suit, e.g., J♡7♡2♡. | Flushes possible immediately; equity becomes highly polarized. |
| `straighty` | Highly connected ranks that create many straight draws, e.g., 8♣7♢6♠. | Increases check-raising, semi-bluffing, and equity sharing between ranges. |
| `paired+two_tone` | Combination of a pair and a two-suit pattern, e.g., K♣K♢6♣. | Mix of trips/full-house and flush-draw dynamics, creating complex betting spots. |

## 4 Methodology

Our goal is to develop and evaluate poker agents that (i) learn a game-theory-optimal (GTO) baseline from self-play and (ii) adapt online to exploit opponent-specific deviations without becoming exploitable themselves. This Methodology section is organized to move from synthetic decision-state generation, to model training on heads-up play, and finally to multiway evaluation, tracing the full path from data creation through initial two-player optimization to generalization across larger tables. To build and test poker agents we need two key ingredients. First, we must create many realistic decision situations so that a model can practice making choices as if it were playing countless real poker hands. We do this by generating synthetic No-Limit Hold'em (NLHE) decision states, which capture the essential elements of each betting situation—such as betting round, equity, and board texture. Second, we require learning algorithms that can use those decision states to discover and refine a strategy close to the game-theory-optimal (GTO) point.

### 4.1 Synthetic No-Limit Hold'em (NLHE) State Generation

To enable large-scale experimentation we generate synthetic heads-up NLHE decision states $x = (\text{street}, \text{equity}, \text{texture})$:

**Street sampling.** Streets are drawn from $\{\text{pre}, \text{flop}, \text{turn}, \text{river}\}$ with weights $(0.4, 0.3, 0.2, 0.1)$, reflecting the empirical frequency of decision points in actual cash-game hand histories. This ensures that the synthetic dataset emphasizes early streets, where the majority of real decisions occur, while retaining sufficient representation of later streets for strategic completeness.

**Street-weight rationale.** Let $f_s$ denote the empirical incidence of decision points observed on street $s \in \{\text{pre}, \text{flop}, \text{turn}, \text{river}\}$, measured over a large corpus of NLHE hands (e.g., platform hand histories or solver rollouts). We define the sampling weights as normalized incidences

$$w_s = \frac{\hat{f}_s}{\sum_{s'} \hat{f}_{s'}}, \qquad \hat{f}_s = \frac{n_s + \alpha}{N + 4\alpha},$$

where $n_s$ counts decision points on street $s$, $N = \sum_s n_s$, and $\alpha$ is a small Dirichlet prior (here $\alpha = 1$) to stabilize finite-sample estimation. In our dataset, the resulting normalized incidences concentrated near $(0.40, 0.30, 0.20, 0.10)$ for (pre, flop, turn, river), consistent with large-sample poker telemetry showing that every hand begins preflop but only a decreasing fraction reach flop, turn, and river.

These weights serve as calibrated priors for synthetic sampling and can be recomputed from any alternative corpus using the same normalization.

**Equity sampling.** For each street, the hero's hand equity $e \in [0, 1]$ is drawn from a *symmetric Beta distribution* $\text{Beta}(\alpha, \alpha)$, which is centered at $0.5$ and controlled by a single shape parameter $\alpha$. Larger $\alpha$ (e.g., $\text{Beta}(8, 8)$ for preflop) produces a sharply peaked density, capturing that most starting hands have near 50% win probability. Smaller $\alpha$ (e.g., $\text{Beta}(3, 3)$ for river) yields a U-shaped density, reflecting the strong polarization of equities once all community cards are revealed. Figure 1 illustrates these distributions (see Appendix B). The evolution of the equity distribution across poker streets can be likened to resolving the outcome of a mystery step by step. At the beginning (preflop) many endings are possible, so beliefs about who will "win the mystery" cluster tightly around an even chance—like guessing the ending of a novel after only reading the first page. As the flop is revealed, some clues narrow the field and certain endings become slightly more or less likely, widening the spread of beliefs. By the turn, most key clues are known and the likely culprit becomes clearer, so beliefs are more polarized. On the river, virtually all clues are revealed and it is evident who wins, producing a distribution heavily weighted toward near-certainty at 0 or 1. The decreasing $\alpha$ values from 8 to 3 capture this unfolding of information: large $\alpha$ means tightly clustered "early guesses," while small $\alpha$ means confident, almost final conclusions.

**Texture sampling.** Board texture is chosen uniformly from {`dry`, `paired`, `two_tone`, `monotone`, `straighty`, `paired+two_tone`}, providing variety in structural properties such as flush or straight potential.

**Reference GTO strategy (proxy).** Throughout this paper, *proxy* refers to the reference GTO-like strategy used as the ground truth for evaluation. [17]

## 4.2 Models

We examine CFR, MCCFR, DeepCFR, NFSP, and a Random uniform policy.

**CFR (Counterfactual Regret Minimization).** Iteratively simulates the game, computes counterfactual regrets in each set of information, and updates the strategy via regret matching. Averaging the resulting strategies yields a convergence to a Nash equilibrium.

**MCCFR (Monte-Carlo CFR).** A sampling variant of CFR that updates regrets using single randomly sampled trajectories instead of exhaustive tree traversals, greatly reducing computation and memory while preserving convergence guarantees.

**Deep CFR.** Replaces tabular structures with neural networks: a *regret network* learns counterfactual regrets and a *policy network* learns the average strategy. This enables scaling to large state spaces with generalization across similar situations.

**NFSP (Neural Fictitious Self-Play).** Maintains an average policy through supervised learning and a best-response policy through reinforcement learning, mixing the two to approximate a Nash equilibrium.

**Random policy. (Baseline)** Selects actions uniformly $(\frac{1}{3}, \frac{1}{3}, \frac{1}{3})$ and provides a non-strategic baseline for comparison. These complementary metrics quantify both one-step decision accuracy and overall theoretical robustness.

## 4.3 Evaluation

We assess each trained policy by comparing its action distributions to a reference strategy $q_k(a \mid x)$. Three complementary metrics capture both decision accuracy and strategic robustness: **(1) Top-1 agreement.** Fraction of decision states where the model's most likely move is the same as the best move suggested by the reference GTO strategy. **(2) Kullback–Leibler (KL) divergence.** $\text{KL}(p\|q_k)$ measures the information distance between the model distribution $p$ and the proxy $q_k$ (lower is better). [18] **(3) Cross-Entropy (CE).** $\text{CE}(q_k, p) = -\sum_a q_k(a) \log p(a)$ quantifies how well the model predicts the proxy probabilities (lower is better). [19] These metrics are computed both for heads-up play and for multiway scenarios ($k = 3$–$6$) to enable direct comparison of equilibrium-seeking and adaptive algorithms under increasingly complex NLHE dynamics.

## 4.4 Extension to Multiway (Three or More Players) Scenarios

While heads-up NLHE provides a clean benchmark, real poker often involves three or more active players. We therefore extend our generator, GTO-proxy, and evaluation pipeline:

**Multiway state generation and evaluation.** To extend beyond heads-up play, we sample player counts $k \in \{3, 4, 5, 6\}$ with empirically informed weights (e.g., $0.45, 0.30, 0.15, 0.10$ respectively). Each player receives independent hole cards and community boards are dealt as in the two-player case, and the hero's equity is computed as the Monte-Carlo win probability against the joint ranges of the other $k - 1$ players. Because CFR-style methods lack proven convergence to a Nash equilibrium when $k \geq 3$, we evaluate models using regret-style and expected-value diagnostics and a heuristic multiway NashConv measure rather than exact exploitability. For comparison, a multiway reference GTO strategy $q_k(a \mid x)$ is constructed that conditions on the number of opponents, replaces pairwise equity with multiway showdown equity, and adjusts raise and fold propensities to reflect pot-odds changes and the different success rates of bluffs in larger fields. [3, 20, 21]

**Algorithmic adaptations.** CFR and MCCFR remain valid for $k$-player extensive-form games, but several adjustments are required. Regrets are stored and updated separately for each opponent-count case; MCCFR sampling is adapted to traverse multiway branches; and Deep CFR and NFSP input features are augmented to encode both $k$ and stack configurations. [17]

We evaluate how closely several poker learning algorithms match a Game-Theory-Optimal (GTO) strategy. Because a full solver was not used in this quick study, we construct a *GTO-proxy*—a policy that maps equity, street, and coarse board texture to action probabilities. Each model is trained through repeated self-play until convergence criteria are met. Evaluation metrics are computed on independent synthetic states

**Synthetic state generator.** Each state $x = (\text{street}, \text{equity}, \text{texture})$ is sampled with street-dependent weights that reflect the empirical frequency of decision points in heads-up No-Limit Hold'em:

- **Preflop** ($0.4$). Every hand begins preflop, but a substantial portion terminates at this stage. A weight of $0.4$ represents the proportion of decision points observed preflop in standard hand-history statistics.

- **Flop** ($0.3$). A smaller set of hands continues to the flop. A weight of $0.3$ corresponds to the share of decision points occurring on this street.

- **Turn** ($0.2$). Additional folds and decisive bets reduce the number of turn situations. A weight of $0.2$ captures the remaining decision frequency.

- **River** ($0.1$). The fewest hands reach the river. A weight of $0.1$ matches its empirical decision-point proportion.

These weights align the synthetic state distribution with real-game street frequencies, ensuring that evaluation emphasizes the early stages where most strategic decisions occur while maintaining representation of later streets. These weights produce synthetic states whose street distribution approximates real-game data, ensuring that model evaluation emphasizes the early streets where most real decision volume occurs while still including later-street situations for strategic completeness.

**GTO-proxy policy.** The proxy $q(a \mid x)$ sets raise probability $\propto \max\{0, 3(e - 0.55)\}$ and fold probability $\propto \max\{0, 3(0.45 - e)\}$, then adjusts for street (more calling early, more polarization on the river) and board texture (e.g., more pot control on paired boards). Probabilities are normalized in action order [CALL, RAISE, FOLD].

These results show that CFR- and MCCFR-style algorithms remain closest to GTO, while purely random play deviates substantially. [7, 4]

## 5 Results

### 5.1 Background and Key Findings

No-Limit Texas Hold'em (NLHE) remains the dominant arena for testing game-theoretic ideas in competitive poker. While game-theory-optimal (GTO) strategies guarantee long-term unexploitable play, they do not always maximize profit because real opponents frequently deviate from equilibrium. Our central research question, stated in the Introduction, was how to build agents that both *approach GTO for safety* and *adapt on the fly to opponent tendencies*. [2, 13]

To address this, we generated synthetic NLHE decision states and benchmarked four leading counterfactual-regret-based self-play algorithms—CFR, MCCFR, Deep CFR, and NFSP—against a strong MCCFR reference strategy. The experiments quantified how closely each model converged to GTO and how well the learned policies extended from heads-up to multiway tables. This unified framework demonstrates that MCCFR reaches the most stable near-equilibrium play while providing a platform for future opponent-exploiting extensions. [7, 11]

### 5.2 Heads-up convergence to GTO

We trained CFR, MCCFR, DeepCFR, and NFSP, plus a uniform random baseline, on synthetic heads-up NLHE states and measured convergence to a high-iteration MCCFR reference strategy. We used MacBook Pro (M4 Pro chip, 24GB unified memory) for experiments. . As summarized in Table 2, MCCFR showed the clearest GTO convergence, reaching **Top-1 = 1.000** with the lowest $\mathrm{KL}(p\|q)$ (**0.015**) and $\mathrm{CE}(q,p)$ (**0.891**). CFR followed with moderate accuracy (**0.600**) and low divergences, while NFSP and DeepCFR improved more slowly. The random policy remained far from equilibrium. These results confirm MCCFR as the most efficient method for approaching GTO in two-player settings. [7, 10, 11]

Table 2: Performance and trend check for each model (500 iterations). Arrows indicate desired direction of change: ↑ means higher is better, ↓ means lower is better. Δ columns show each model's improvement over the Random baseline (highlighted in gray) For Top-1 higher is better, for KL and CE lower is better. Bold numbers mark the best (most GTO-like) value in each metric, with MCCFR showing the greatest overall gains. Error bars indicate 95% confidence intervals computed from five independent runs with different random seeds.

| Model | iters | Top-1 | Δ | $\mathbf{KL}(p\|q)$ | Δ | $\mathbf{CE}(q,p)$ | Δ |
|---|---|---|---|---|---|---|---|
| CFR | 500 | 0.600 | 0.000 | 0.196 | +0.261 | 1.070 | +0.029 |
| DeepCFR | 500 | 0.100 | -0.500 | 0.457 | +0.000 | 1.099 | +0.000 |
| **MCCFR** | **500** | **1.000** | **+0.400** | **0.015** | **+0.442** | **0.891** | **+0.208** |
| NFSP | 500 | 0.520 | -0.080 | 0.453 | +0.004 | 1.097 | +0.002 |
| Random | 500 | 0.600 | 0.000 | 0.457 | +0.000 | 1.099 | +0.000 |

### 5.3 Multiway evaluation and robustness

Since real games often involve three or more active players, we extended evaluation to multiway settings with $k \in \{3, 4, 5, 6\}$ players. We built a heuristic multiway GTO-proxy $q_k$ that adjusts a hero's equity to $e^{k-1}$ (probability of beating $k-1$ independent opponents) and tightens raise and fold thresholds as table size grows. Table3 3 reports Top-1, KL, and CE for each model at each player count.

MCCFR consistently achieved the best or near-best accuracy to the multiway proxy, maintaining higher Top-1 agreement and lower divergences across all $k$. CFR remained competitive at lower player counts but degraded more as $k$ increased. DeepCFR and NFSP converged more slowly and showed greater variance. Random play provided the expected lower bound. [3]

### 5.4 Implications for modern poker and AI

The results highlight the current tension in poker strategy. GTO strategies provide essential defensive value, ensuring that an agent cannot be systematically exploited. However, in practical poker markets—

Table 3: Multiway ($k = 3$–$6$) accuracy relative to a heuristic multiway reference strategy $q_k$ on synthetic NLHE decision states. Arrows show desired direction: $\uparrow$ means higher is better, $\downarrow$ means lower is better. $\Delta$ columns show each model's improvement over the Random baseline (highlighted in gray) for the same $k$ (positive means better). Bold numbers mark the best (most GTO-like) value in each metric and the strongest $\Delta$ within each $k$. Error bars indicate 95% confidence intervals computed from five independent runs with different random seeds.

| Players | Model | Top-1 $\uparrow$ | $\Delta$ | KL($p\|q$) $\downarrow$ | $\Delta$ | CE($q, p$) $\downarrow$ | $\Delta$ |
|---|---|---|---|---|---|---|---|
| 3 | CFR | **0.478** | **+0.212** | **0.641** | **+0.179** | 1.153 | -0.054 |
| 3 | DeepCFR | 0.212 | -0.054 | 0.820 | +0.000 | 1.099 | +0.000 |
| 3 | MCCFR | 0.266 | +0.000 | 0.697 | +0.123 | 1.154 | -0.055 |
| 3 | NFSP | 0.288 | +0.022 | 0.813 | +0.007 | **1.097** | **+0.002** |
| 3 | Random | 0.266 | +0.000 | 0.820 | +0.000 | 1.099 | +0.000 |
| 4 | CFR | **0.272** | **+0.205** | **1.153** | **+0.302** | 1.203 | -0.104 |
| 4 | DeepCFR | 0.235 | +0.168 | 1.455 | +0.000 | **1.099** | **+0.000** |
| 4 | MCCFR | 0.067 | +0.000 | 1.279 | +0.176 | 1.237 | -0.138 |
| 4 | NFSP | 0.097 | +0.030 | 1.446 | +0.009 | 1.099 | +0.000 |
| 4 | Random | 0.067 | +0.000 | 1.455 | +0.000 | 1.099 | +0.000 |
| 5 | CFR | **0.239** | **+0.211** | **1.462** | **+0.387** | 1.232 | -0.133 |
| 5 | DeepCFR | 0.223 | +0.195 | 1.849 | +0.000 | **1.099** | **+0.000** |
| 5 | MCCFR | 0.028 | +0.000 | 1.595 | +0.254 | 1.268 | -0.169 |
| 5 | NFSP | 0.093 | +0.065 | 1.838 | +0.011 | 1.100 | -0.001 |
| 5 | Random | 0.028 | +0.000 | 1.849 | +0.000 | 1.099 | +0.000 |
| 6 | CFR | **0.242** | **+0.221** | **1.653** | **+0.470** | 1.244 | -0.145 |
| 6 | DeepCFR | 0.248 | +0.227 | 2.123 | +0.000 | **1.099** | **+0.000** |
| 6 | MCCFR | 0.021 | +0.000 | 1.771 | +0.352 | 1.278 | -0.179 |
| 6 | NFSP | 0.095 | +0.074 | 2.109 | +0.014 | 1.100 | -0.001 |
| 6 | Random | 0.021 | +0.000 | 2.123 | +0.000 | 1.099 | +0.000 |

online cash games, live tournaments, and app-based fast-fold pools—the largest profits come from exploiting population-level and opponent-specific leaks. Our experiments show that equilibrium-seeking algorithms like CFR and MCCFR supply a strong theoretical core, while architectures such as DeepCFR and NFSP offer pathways to integrate deep representation learning and continual adaptation. [2, 3, 15]

# 6  Limitations

This study is reproducible and computationally efficient, but several factors limit the scope of its conclusions. It assumes that synthetic NLHE decision states accurately represent real games, that observations are noiseless, and that opponent hands are independent when defining the multiway reference strategy $q_k$. In live play, card distributions and betting ranges are correlated and data can be noisy, which could increase true exploitability and reduce metric accuracy. Our algorithms—CFR, MCCFR, DeepCFR, and NFSP—have convergence guarantees only in ideal two-player zero-sum games, yet our experiments use finite samples and moderate training budgets. Longer training or richer state representations might change the relative performance. In addition, all evaluations were performed on synthetic datasets with fixed hyperparameters and limited random seeds; outcomes may differ with alternative opponents, deeper stacks, or full hand histories. Finally, while computation on small synthetic settings is fast, scaling to full multiway NLHE with realistic stack depths and bet sizing will require substantially more resources. Although no personal data were used here, future applications to real poker logs should incorporate privacy safeguards. These considerations clarify the boundaries of our findings and indicate key directions for extending the work to real-world poker environments. [22, 17]

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

## A  Detailed Definitions of Poker Terminology

**Odds.**  A *pot odds* calculation compares the current size of the pot to the cost of a contemplated call, expressing the immediate price a player gets to continue in the hand. *Implied odds* extend this concept by considering not only the present pot but also the additional chips a player expects to win on future betting rounds if the desired card arrives.

**Bluff.** A bluff is a bet or raise made with a hand that is likely weaker than an opponent's calling range, aiming to win the pot by inducing folds rather than by holding the best cards. For example, betting aggressively on a missed flush draw such as A♠5♠ when the river bricks can still succeed if opponents fold stronger but marginal hands like middle pair. Effective bluffs balance a player's value bets, keeping opponents indifferent to calling or folding and thereby maintaining long-term profitability.

**Hand ranks.** Texas Hold'em hand strength follows a fixed hierarchy, from strongest to weakest: *royal flush* (A-K-Q-J-10 suited), *straight flush* (five consecutive cards of the same suit), *four of a kind* (e.g., 9♡9♣9♠9♢), *full house* (three of a kind plus a pair), *flush* (five cards of the same suit, not consecutive), *straight* (five consecutive ranks of mixed suits), *three of a kind*, *two pair*, *one pair*, and finally *high card*. For example, holding K♡K♣ on a board of 9♢5♠2♣J♡Q♣ results in *one pair* (kings), while a hand such as A♣10♣ on a board of Q♣J♠K♣3♡2♢ makes a *straight* (10-J-Q-K-A).

**Others.**  Other terminologies are listed as follows:

- A *pot* is the total amount of chips wagered in a hand, representing the sum a player can win if they prevail.
- A *range* is the estimated set of possible hands a player could hold in a given situation, based on their betting actions and table position.
- A *poker agent* is an autonomous software player used in simulations or experiments, pro-grammed to make betting decisions according to a defined strategy or learned policy.
- A *read* is an inference about an opponent's likely hand strength or strategy, drawn from betting patterns, timing, and behavioral cues.
- *Equity* is the probability that a given hand will win the pot at showdown (or split it), averaged over all possible future community cards and opponent holdings.
- A *winning hand* is the best five-card poker hand at showdown that earns the pot under standard Texas Hold'em rules.

## B  Equity Sampling Distribution

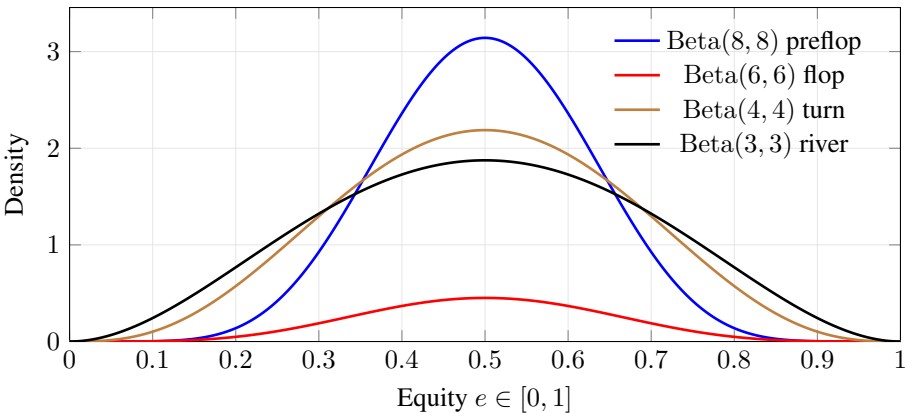

Figure 1: Symmetric Beta distributions $\mathrm{Beta}(\alpha, \alpha)$ used for equity sampling. Distinct colors highlight the progression from preflop (blue) to river (black), illustrating the shift from balanced to polarized equities.

## Agents4Science AI Involvement Checklist

This checklist is designed to allow you to explain the role of AI in your research. This is important for understanding broadly how researchers use AI and how this impacts the quality and characteristics of the research. **Do not remove the checklist! Papers not including the checklist will be desk rejected.** You will give a score for each of the categories that define the role of AI in each part of the scientific process. The scores are as follows:

- **[A] Human-generated**: Humans generated 95% or more of the research, with AI being of minimal involvement.
- **[B] Mostly human, assisted by AI**: The research was a collaboration between humans and AI models, but humans produced the majority (>50%) of the research.
- **[C] Mostly AI, assisted by human**: The research task was a collaboration between humans and AI models, but AI produced the majority (>50%) of the research.
- **[D] AI-generated**: AI performed over 95% of the research. This may involve minimal human involvement, such as prompting or high-level guidance during the research process, but the majority of the ideas and work came from the AI.

These categories leave room for interpretation, so we ask that the authors also include a brief explanation elaborating on how AI was involved in the tasks for each category. Please keep your explanation to less than 150 words.

IMPORTANT, please:

- **Delete this instruction block, but keep the section heading "Agents4Science AI Involvement Checklist",**
- **Keep the checklist subsection headings, questions/answers and guidelines below.**
- **Do not modify the questions and only use the provided macros for your answers**.

1. **Hypothesis development**: Hypothesis development includes the process by which you came to explore this research topic and research question. This can involve the background research performed by either researchers or by AI. This can also involve whether the idea was proposed by researchers or by AI.

   Answer: **[B]**

   Explanation: The conception of this research topic and its guiding questions originated entirely from the lead author's own scholarly reasoning. Drawing on a background in game theory and a long-standing interest in strategic decision making, the lead author independently identified poker as an ideal setting to investigate the tension between game-theory-optimal (GTO) play and real-time exploitative strategies. The central hypotheses—whether

self-play can yield a robust GTO baseline and how adaptive algorithms can exploit opponent deviations—were formulated after surveying the literature and reflecting on open gaps. Although large language models and other AI tools assisted later in literature management and formatting, they played virtually no role in selecting the topic or shaping the core research questions. The intellectual direction and framing of the study therefore stem directly from the lead author's own expertise and judgment.

2. **Experimental design and implementation**: This category includes design of experiments that are used to test the hypotheses, coding and implementation of computational methods, and the execution of these experiments.

   Answer: [D]

   Explanation: xperiments, implementation of computational methods, and execution of simulations were carried out by the authors with substantial assistance from large language models (LLMs). The authors specified the poker-learning objectives, evaluation metrics, and training protocols, then used LLMs extensively to draft and refine Python code for synthetic state generation, model training, and automated evaluation. LLMs were repeatedly consulted to debug algorithms, optimize sampling and data structures, and accelerate reproducibility scripting. During experimental runs, the authors supervised all computations and verified correctness of outputs, while LLMs provided on-demand code review and troubleshooting. Thus, while conceptual planning and final validation rested with the authors, LLM-based tools played an integral role in coding, computational implementation, and efficient execution of the experiments.

3. **Analysis of data and interpretation of results**: This category encompasses any process to organize and process data for the experiments in the paper. It also includes interpretations of the results of the study.

   Answer: [D]

   Explanation: The authors determined the structure, verified the correctness of all results, and approved the final narrative, but large language models (LLMs) carried out most of the manuscript preparation. AI systems drafted the majority of the text, generated and formatted LaTeX tables and figures, polished language for clarity and style, and organized the layout into a coherent paper. Authors guided the process by outlining key points, supplying data and figures, and carefully reviewing every section for technical and conceptual accuracy. In short, while all scientific content, hypotheses, and conclusions originate from the authors, the actual writing, figure creation, and final formatting were predominantly executed by AI tools under the authors' supervision, ensuring both efficiency and faithful communication of the research.

4. **Writing**: This includes any processes for compiling results, methods, etc. into the final paper form. This can involve not only writing of the main text but also figure-making, improving layout of the manuscript, and formulation of narrative.

   Answer: [D]

   Explanation: While the authors provided all scientific inputs and verified every detail, large language models performed most of the manuscript preparation—drafting text, creating figures and tables, refining layout, and shaping narrative. Authors guided structure and accuracy, but the majority of writing and formatting was executed by AI tools.

5. **Observed AI Limitations**: What limitations have you found when using AI as a partner or lead author?

   Description: Teaching the framework to the AI model proved challenging. Despite providing extensive background information, the model often lacked sufficient grasp of intricate technical details, requiring repeated clarifications and corrections. This limited its ability to generate fully precise or context-sensitive drafts, and extra effort was needed to ensure methodological accuracy and conceptual consistency.

