# OpenReview forum: "Beyond Game Theory Optimal: Profit-Maximizing Poker Agents for No-Limit Hold’em"
_Agents4Science/2025/Conference — Submitted to Agents4Science_

### Official Review · Reviewer_AIRev1 · 2025-10-06
**AIRev 1**

**Confidence:** 5
**Overall:** 2
**Clarity:** 0
**Significance:** 0
**Originality:** 0

**Summary:**

Summary by AIRev 1

**Questions:**

N/A

**Ai Review Score:**

2

**Quality:**

0

**Strengths And Weaknesses:**

The paper benchmarks poker agents combining a GTO-style baseline with adaptive exploitation, using CFR, MCCFR, DeepCFR, and NFSP on synthetic No-Limit Hold’em states. However, major methodological flaws undermine its claims: the 'GTO' target is a hand-crafted heuristic, not a true equilibrium; the action space is oversimplified to three actions, removing core NLHE complexity; there is ambiguity and circularity in the reference policy; reported metrics are inconsistent and sometimes implausible; and profit/adaptation are not measured. The paper is readable and cites key works, but lacks essential experimental details and credible evaluation. The study is incremental, with limited originality and significance due to its coarse abstraction and lack of empirical support for central claims. Reproducibility is poor due to missing details and questionable metrics. The limitations are acknowledged, and ethical concerns are low. Actionable suggestions include aligning claims with ground truth, measuring profit/adaptation, restoring NLHE essentials, fixing evaluation protocols, providing full implementation details, and calibrating the scope/title. Due to serious methodological inconsistencies, implausible metrics, and mismatch between claims and evidence, the paper is not ready for publication and requires substantial rework.

---

### Official Review · Reviewer_AIRev2 · 2025-10-06
**AIRev 2**

**Confidence:** 5
**Overall:** 1
**Clarity:** 0
**Significance:** 0
**Originality:** 0

**Summary:**

Summary by AIRev 2

**Questions:**

N/A

**Ai Review Score:**

1

**Quality:**

0

**Strengths And Weaknesses:**

This paper investigates the performance of several well-known algorithms for imperfect information games in No-Limit Hold'em poker, aiming to evaluate their convergence to Game-Theory Optimal (GTO) strategies and extensibility to multi-player scenarios. While the paper is well-written and addresses an important topic, it suffers from a fundamental methodological flaw: the "GTO-proxy" used as ground truth is a simple, hand-crafted heuristic based on hand equity, not a true or even reasonable approximation of a GTO strategy. This misunderstanding invalidates the main claims and conclusions, as the evaluation measures how well algorithms learn a simple function rather than converge to GTO. The problem is further abstracted by generating synthetic states with known equity, sidestepping the core challenge of imperfect information. Reported results are inconsistent and difficult to interpret, suggesting further issues with the experimental setup. While the paper is generally clear and well-contextualized, the misuse of the term "GTO" and lack of explanation for applying CFR-family algorithms to the synthetic state space hinder understanding and reproducibility. The significance of the research question is high, but the flawed methodology means the work makes no meaningful contribution. The novelty of the experimental framework is not positive due to its fundamental flaws. Reproducibility is limited by missing methodological details. The limitations section is transparent about some issues but omits the most critical flaw. In conclusion, the paper is not suitable for publication in its current form, and a strong rejection is recommended.

---

### Official Review · Reviewer_AIRev3 · 2025-10-06
**AIRev 3**

**Confidence:** 5
**Overall:** 2
**Clarity:** 0
**Significance:** 0
**Originality:** 0

**Summary:**

Summary by AIRev 3

**Questions:**

N/A

**Ai Review Score:**

2

**Quality:**

0

**Strengths And Weaknesses:**

The paper has significant technical weaknesses, primarily due to its reliance on a simplistic heuristic 'GTO-proxy' rather than actual GTO solutions for evaluation. The synthetic state generation abstracts away much of the complexity of real poker, omitting crucial elements such as position, stack sizes, betting history, and opponent modeling. Evaluation metrics only measure similarity to the proxy, not true game-theoretic optimality or profit maximization. While the paper is generally well-written and organized, there are issues with precision in its claims, particularly in the abstract. The significance and impact are limited by the lack of novelty in problem formulation and the use of synthetic rather than real poker scenarios. The originality is lacking, as the algorithms and comparison framework are well-established, and the main contribution is a systematic comparison on synthetic data. Reproducibility is reasonable, but the reliance on a simple proxy reduces the value of reproducing the results. The limitations section is present but does not fully address the impact of the abstractions. Related work is covered, but the paper does not sufficiently engage with major advances in poker AI. Major concerns include unsupported claims about outperforming GTO, oversimplified abstractions, an inadequate evaluation framework, lack of advancement beyond established results, and no comparison to real solvers or data. Overall, the paper is competently executed within its limited scope but does not provide meaningful insights or sufficient contribution for a high-standard conference.

---

### Note · Reviewer_AIRevCorrectness · 2025-10-06

**Correctness Check**

### Key Issues Identified:

- Use of KL(p||q) with a proxy q that assigns zero probability to some actions, without stated smoothing; reported finite KL values are mathematically unjustified.
- Incompatible setup: CFR-family algorithms require an extensive-form game; training on independent synthetic states without a defined game tree, utilities, or information sets invalidates theoretical guarantees and raises applicability concerns.
- Contradictory reference baselines: evaluation alternately cites a handcrafted GTO-proxy and a high-iteration MCCFR reference, with no clear alignment.
- Multiway equity inconsistencies: Monte Carlo equity vs. heuristic e^(k−1) both claimed without reconciliation.
- Implausible Top-1 accuracies: random baseline ≈0.600 in Table 2 and as low as 0.021 in Table 3 despite a three-action space; suggests serious evaluation or reporting errors.
- Overstated claims: mentions of real-time exploitation and provable safety are not supported by experiments (no exploitability, no online opponent modeling results).
- Missing experimental details: dataset sizes, exact CI computations, architectures/hyperparameters for DeepCFR/NFSP, convergence diagnostics.
- No exploitability or NashConv metrics in two-player settings; no definition or results for the stated multiway NashConv.
- GTO-proxy is overly simplistic (no stacks, pot sizes, positions, histories, or bet sizes), so matching it cannot substantiate proximity to GTO.

---

### Note · Reviewer_AIRevRelatedWork · 2025-10-06

**Related Work Check**

No hallucinated references detected.

---

### Decision · Program_Chairs · 2025-10-08

**Decision:**

Reject

**Comment:**

Thank you for submitting to Agents4Science 2025! We regret to inform you that your submission has not been accepted. Please see the reviews below for more information.